# Understanding the Design Principles of Link Prediction in Directed Settings

## Abstract

Link prediction is a widely studied task in *Graph Representation Learning (GRL)* for modeling relational data. Early theories in GRL were based on the assumption of a symmetric adjacency matrix, reflecting an undirected setting. As a result, much of the following state-of-the-art research has continued to operate under this symmetry assumption, even though real-world data often involves crucial information conveyed through the direction of relationships. This oversight limits the ability of these models to fully capture the complexity of directed interactions. In this paper, we focus on the challenge of directed link prediction by evaluating key heuristics that have been successful in the undirected settings. We propose simple but effective adaptations of these heuristics to the directed link prediction task and demonstrate that these modifications yield competitive performance compared to leading *Graph Neural Networks (GNNs)* originally designed for undirected graphs. Through an extensive set of experiments, we derive insights that inform the development of a novel framework for directed link prediction, which not only surpasses baseline methods but also outperforms state-of-the-art GNNs on multiple benchmarks.

## 1    Introduction

Link prediction is a task that seeks to uncover missing connections (*e.g.* links), between entities (*e.g.* vertices) in a graph, and has many industrial applications. For example, an e-commerce platform might represent their users and items as vertices, and the transactions users make as edges from user to item. In this setting, a recommendation system that predicts items that users might like to buy can readily be cast as a link prediction task (Chamberlain et al., 2022; Wang et al., 2023). Beyond direct applications, link prediction is often used in unsupervised settings to construct vertex representations that can then be used in various downstream tasks, such as fraud or toxicity detection (Pal et al., 2020; El-Kishky et al., 2022; Liu et al., 2020; Zhang et al., 2022).

Various methodologies for link prediction have been developed and can be broadly classified into three categories. The first category, *similarity-based heuristics*, involves computing a score for each pair of nodes to quantify their similarity (Wang et al., 2007). These scores are then ranked, with higher scores indicating a greater likelihood of connection between node pairs. The second category encompasses *probabilistic and maximum likelihood models* (Wang et al., 2007; Clauset et al., 2008; Guimerà & Sales-Pardo, 2009). Although these models have demonstrated effectiveness on smaller datasets, they tend to be computationally intensive and face scalability challenges in large real-world graphs. The third category includes *node representation learning methods*, where an *encoder* learns to represent each node as a vector in an embedding space, and a *decoder* processes pairs of node embeddings to generate a score that quantifies the likelihood of a link existing. These embeddings are optimized so that nodes with similar neighborhood structures are represented similarly in the embedding space. Node representation methods can be further divided into three subcategories based on the choice of encoder: *random walk-based approaches* (Perozzi et al., 2014; Trouillon et al., 2016; Cao et al., 2018; Kazemi & Poole, 2018), *matrix decomposition techniques* (Acar et al., 2009; Kazemi & Poole, 2018), and *Graph Neural Networks (GNNs)* (Kipf & Welling, 2016; Hamilton et al., 2017; Ying et al., 2018; Veličković et al., 2017).

Although link prediction is valuable across many applications, most widely used methods assume that links are undirected. In many settings, this assumption makes sense. For example, an edge

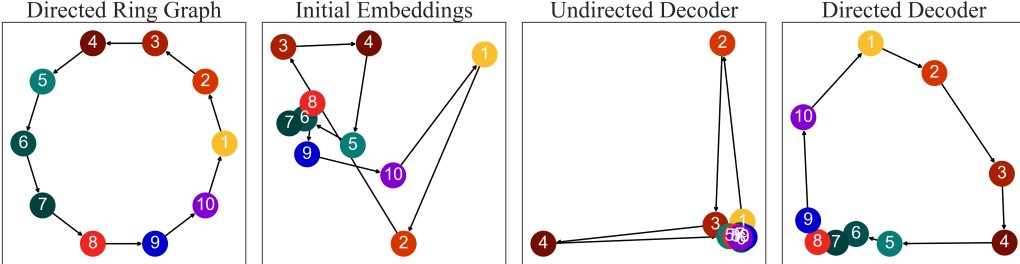

Figure 1: **An illustrative evaluation demonstrating the impact of incorporating directionality into the design of predictive models.** We generate a directed ring graph $\mathcal{G} = (\mathcal{V}, \mathcal{E})$. Each node $u \in \mathcal{V}$ is initialized with a two-dimensional embedding $\mathbf{e}_u^{(0)}$. We then train GraphSage (Hamilton et al., 2017), to update these embeddings using two different decoders: (1) a undirected decoder, and (2) a directed decoder, to perform link prediction. By visualizing the output node embeddings in both cases, we observe that the structural representation of nodes is significantly enhanced when using a directed decoder, emphasizing the importance of directionality in the model design.

in the Facebook friendship graph will be undirected, because friendships are bidirectional on the platform. As a result, it is common to transform directed graphs into undirected ones. However, this undirected transformation does not make sense in all settings because an edge's directionality can denote the semantics of that edge, which is often crucial for accurately modeling interactions.

Intuitively, directed edges allow us to differentiate a node's role as either a source or a target within a network, which provides finer granularity when modeling interactions. In many real-world contexts, this asymmetry is meaningful. For instance, consider a transaction network where the goal is to identify fraudulent behavior. Suppose there are three types of participants: a parent, a teenager, and a fraudster. Let's assume the model aims to understand the identity and behavior of the teenager. A money transfer from a parent to a teenager represents one type of pattern, while a transfer from a teenager to a fraudster at the same monetary amount represents another. If we look at a scenario where the teenager is receiving money from the parent, this pattern might be considered safe—such as receiving an allowance. However, if the teenager is instead sending the same amount to a fraudster, this could be a red flag. By taking into account the direction of these transactions, we can more effectively distinguish between benign and suspicious patterns. In a directed setting, the directionality conveys vital information about who initiates and who receives a transaction, which can significantly impact downstream predictions. Conversely, if we convert the originally directed network into an undirected one by symmetrizing the adjacency matrix, the sender/receiver roles of the teenager in these transactions would be lost, potentially leading to a misinterpretation of the interactions and harming predictive performance.

To make this point clear, we have constructed a directed ring graph and present it in Figure 1, as well as embeddings generated using both an undirected and a directed graph decoder. We clearly observe that the asymmetric, or directed, decoder is more capable of replicating the expected planar positioning, while the undirected decoder generates a set of embeddings where almost all vertices collapse together in the embedding space. This example helps us to develop the intuition that using undirected link predictors in directed settings may lead to poor performance.

In this paper, we take this intuition and make it concrete by exploring the problem of directed link prediction. Along the way, we gain insights into the design principles needed for the development of link predictors that are applicable to directed settings. The key contributions of our work are as follows:

- We establish **a robust comparison framework** by constructing three types of baseline models: heuristic approaches, Multi-Layer Perceptron (MLP), and GNN based models and their variants for directed settings.

- We conduct extensive ablation experiments to extend the known **design principles** of link prediction to include their directed variants, and study the impact of directionality on the predictive performance for each such design principle.

- Based on the design principles determined, we develop DirLP, **a novel model for directed link prediction** that significantly outperforms the baseline models.

This work not only establishes a new state-of-the-art in directed link prediction but also offers actionable insights that inform the design of future models. Our contributions aim to bridge the gap between undirected and directed link prediction research, providing strong foundations for future work in this important area.

## 2 RELATED WORK

For the review of literature, we focus on *similarity-based heuristics* for link prediction, due to their practical applicability, and *Graph Neural Networks (GNNs)*, which represent the state-of-the-art in the field.

**Similarity-based Heuristics.** A variety of similarity-based heuristics have been developed for the task of link prediction, primarily focusing on quantifying node similarity to predict the likelihood of a connection. One of the earliest and simplest methods is the *Common Neighbors (CN)* heuristic, where the number of shared neighbors between two nodes is used as an indicator of their likelihood to form a link. Extensions of this idea include the *Jaccard Index (JI)* and *Adamic-Adar index (AA)*, which provide weighted variations by considering the degree of shared neighbors Adamic & Adar (2003); Liben-Nowell & Kleinberg (2007). The *Preferential Attachment (PA)* heuristic, based on the idea that high-degree nodes are more likely to attract additional links, is another widely used method Newman (2001). *Local Path index (LP)* (Lü et al., 2009) expands upon the idea of common neighbors by considering paths of length two between node pairs. It balances between global and local information, providing a broader view of node similarity while still being computationally feasible for large graphs. *Resource Allocation index (RA)* (Zhou et al., 2009) is another similarity-based measure, where the likelihood of a link is determined by how resources (or connections) are shared between two nodes via their common neighbors. It gives higher weight to common neighbors with lower degrees, assuming that connections from lower-degree nodes are more significant. While these heuristics are computationally efficient and effective in many settings, they are primarily designed for undirected graphs and tend to struggle when applied to directed graphs. These methods also assume that local structural properties of the graph are sufficient for prediction, limiting their ability to capture more complex relational patterns. Despite these limitations, similarity-based heuristics remain popular due to their simplicity and interpretability, often serving as strong baselines for more advanced models like GNNs.

**Graph Neural Networks (GNNs).** Most of the popular GNNs (Kipf & Welling, 2016; Hamilton et al., 2017; Veličković et al., 2017; Ying et al., 2018) primarily focus on node representation in undirected graphs. Several studies have specifically addressed various aspects of directed graphs. For example, GatedGCN (Li et al., 2015), which employs separate aggregations for in-neighbors and out-neighbors in directed graphs, has proven effective for solving the genome assembly problem (Vrček et al., 2022). Additionally, research has aimed to generalize spectral convolutions for directed graphs Ma et al. (2019); Monti et al. (2018); Tong et al. (2020b;a). A notable contribution is made by Zhang et al. (2021b), who present a spectral method that utilizes a complex matrix for graph diffusion, where the real part represents the undirected adjacency and the imaginary part captures the edge direction. Building on their work, Geisler et al. (2023) proposed a positional encoder that integrates transformers into directed graphs. More recently, Rossi et al. (2024) emphasized that effective link prediction in directed graphs necessitates distinct aggregation strategies for incoming and outgoing edges to fully leverage directional information. They proposed a novel and generic *Directed Graph Neural Network (Dir-GNN)* that can be integrated with any message-passing neural network by implementing separate aggregations of incoming and outgoing edges.

Once the learned node embeddings are obtained, the link prediction problem can be framed as a supervised binary classification task. In this context, the input consists of a pair of node embeddings corresponding to the link of interest, while the output is a score that quantifies the probability of the existence of that link. Various decoders have been proposed to achieve this classification, each with distinct methodologies. One of the simplest and most widely used decoders is the dot product decoder (Kipf & Welling, 2016). However, this method fails to account for the directionality of links since the dot product is commutative. For instance, in a transactional context, the likelihood of person A transferring money to person B is treated the same as that of person B transferring money

to person A, which hinders accurate predictions of money flow. To address the limitations of the dot product, Bilinear Decoders introduce a learned weight matrix, providing a more nuanced approach to edge prediction (Yang et al., 2014). Another method for quantifying the similarity between two nodes is the distance-based decoder, which predicts edge existence based on the distance between node embeddings (Ou et al., 2016). Matrix factorization-based decoders decompose the adjacency matrix into low-rank matrices representing node embeddings. The reconstructed adjacency matrix can then be used for link prediction (Tang et al., 2015). Neural Tensor Network (NTN) (Socher et al., 2013) replaces the standard linear layer with a bilinear tensor layer, directly relating the two nodes. A novel decoder inspired by Newton's theory of universal gravitation was introduced by Salha et al. (2019). This approach uses node embeddings to reconstruct asymmetric relationships, facilitating effective directed link prediction.

## 3 BACKGROUND AND PRELIMINARIES

In this section we introduce the notation employed throughout the paper in addition to metrics and datasets used for experiments. Next, we provide an overview of two main frameworks adapted to address directed link prediction problem in our analysis: (1) similarity-based heuristics and (2) Graph Neural Networks (GNNs).

**Notation.** Given a graph $\mathcal{G} = (\mathcal{V}, \mathcal{E})$ where $\mathcal{V}$ and $\mathcal{E}$ denote the set of vertices and edges respectively, *directed link prediction* refers to the task of predicting the existence of an edge *from $u \in \mathcal{V}$ to $v \in \mathcal{V}$*. The adjacency matrix of $\mathcal{G}$ is denoted by $\mathbf{A}$, where $\mathbf{A}_{uv} = 1$ if there is a directed edge from node $u$ to $v$, and $\mathbf{A}_{uv} = 0$, otherwise. Additionally, we denote the neighbourhood of a given node $u$ as $\mathcal{N}(u)$ throughout the paper. Each node $u \in \mathcal{V}$ is associated with a feature vector $\mathbf{x} \in \mathbb{R}^d$ where $d$ is the feature dimensionality.

**Metrics.** In general, to evaluate the prediction performance of a given method, the set of edges is divided into disjoint sets of training and testing splits; $\mathcal{E}_{\text{train}}$ and $\mathcal{E}_{\text{test}}$, respectively. In this paper, we evaluate the predictive performance mainly by mean reciprocal rank (MRR), which is calculated as the average of the reciprocal ranks of the true positives in the test set. The MRR is formulated as follows:

$$\text{MRR} = \frac{1}{|\mathcal{E}_{\text{test}}|} \sum_{(u,v) \in \mathcal{E}_{\text{test}}} \frac{1}{\text{rank}(u,v)}, \tag{1}$$

where $\text{rank}(u,v)$ is the rank of the true link $(u,v)$ among possible candidate links involving node $u$.

**Datasets.** In our experiments, we evaluate directed link prediction performance of various approaches using six benchmark datasets: CORA (Yang et al., 2016), CITESEER (Yang et al., 2016), CHAMELEON (Rozemberczki et al., 2021), SQUIRREL (Rozemberczki et al., 2021), BLOG (He et al., 2022), WIKICS (Mernyei & Cangea, 2020). All datasets are directed and come from the PYTORCH GEOMETRIC SIGNED DIRECTED software package (He et al., 2024). We use the directed version of CORA and CITESEER and not their commonly used undirected versions. The details regarding datasets are provided in Appendix A.

### 3.1 SIMILARITY-BASED HEURISTICS

In general, similarity-based heuristic methods used for link prediction assign a similarity score $S(u,v)$ for each pair of nodes, such that $S(u,v)$ serves as an estimator on the likelihood of a link between node $u$ and node $v$. For example, the Resource Allocation (RA) heuristic score between node $u$ and $v$ is calculated as follows:

$$S_{\text{RA}}(u,v) = \sum_{t \in \mathcal{N}(u) \cap \mathcal{N}(v)} \frac{1}{|\mathcal{N}(t)|}. \tag{2}$$

By definition, RA is a symmetric score function, i.e. $S_{\text{RA}}(u,v) = S_{\text{RA}}(v,u)$. In order to adapt the existing similarity scores into directed settings, we utilize *directed neighborhood operator*. Let $\mathcal{N}_{\text{in}}(u)(\mathcal{N}_{\text{out}}(u))$ consists of all nodes that have a directed edge pointing toward (originating from) node $u$. Formally:

$$\mathcal{N}_{\text{in}}(u) = \{v \in \mathcal{V} \mid (v,u) \in \mathcal{E}\}, \ \mathcal{N}_{\text{out}}(u) = \{v \in \mathcal{V} \mid (u,v) \in \mathcal{E}\}. \tag{3}$$

Using $\mathcal{N}_{\text{in}}(\cdot)$ and $\mathcal{N}_{\text{out}}(\cdot)$, one can define four variants of common neighbourhood for a given node pair $u$ and $v$ by $\mathcal{N}_{d_u}(u) \cap \mathcal{N}_{d_v}(v)$ where $d_u, d_v \in \{\text{in}, \text{out}\}$. For example, given the set of nodes that has an incoming link from $u$ and has an outgoing link to $v$, the corresponding RA would be calculated as follows:

$$S_{\text{RA,in-out}}(u,v) = \sum_{t \in \mathcal{N}_{\text{in}}(u) \cap \mathcal{N}_{\text{out}}(v)} \frac{1}{|\mathcal{N}(t)|}. \tag{4}$$

More details regarding the individual methods used in our experiments are provided in Appendix B.

## 3.2 GRAPH NEURAL NETWORKS (GNNs)

**Encoders.** Most common form of encoder used in GNNs is *Message Passing Neural Networks (MPNNs)* in which vector-based messages are passed between nodes and then updated through neural networks to generate node embeddings. MPNNs initialize a set of the hidden node embeddings $\mathbf{h}_u^{(0)}, \forall u \in \mathcal{V}$ by input features, i.e., $\mathbf{h}_u^{(0)} = \mathbf{x}_u$. At the $k^{\text{th}}$ iteration of message passing hidden embeddings $\mathbf{h}_u^{(k)}$ for each node $u \in \mathcal{V}$ are updated as follows:

$$\mathbf{h}_u^{(k+1)} = f_{\text{update}}^{(k)}\left(\mathbf{h}_u^{(k)}, f_{\text{aggregate}}^{(k)}\left(\left\{\mathbf{h}_v^{(k)}, \forall v \in \mathcal{N}(u)\right\}\right)\right), \tag{5}$$

where $f_{\text{update}}(\cdot)$ and $f_{\text{aggregate}}(\cdot)$ are choice of differentiable functions. The final hidden embeddings are used as the output node embeddings $\mathbf{e}_u^{(0)}, \forall u \in \mathcal{V}$:

$$\mathbf{e}_u = \mathbf{h}_u^{(K)}, \forall u \in \mathcal{V}, \tag{6}$$

where $K$ denotes the total number of update layers. In the functional form, an MPNN can be summarized as follows:

$$f_{\text{MPNN}} : \mathcal{G} = (\mathcal{V}, \mathcal{E}) \text{ and } \mathbf{x}_u, \forall u \in \mathcal{V} \longmapsto \mathbf{e}_u, \forall u \in \mathcal{V}. \tag{7}$$

In terms of link prediction task, MPNNs serve as an encoder that maps structural information of the graph together with node features into a set of node embeddings.

**Decoders.** In the context of link prediction, the decoder is defined as $f : \mathbb{R}^{D \times 2} \to \mathbb{R}^+$, which maps the embeddings for a given link to an individual score that corresponds to its likelihood to exist. For edge-wise link-prediction it is common to use one of two link predictors: the dot product (DP) $f_{\text{DP}}(v_i, v_j) = \sigma(\mathbf{e}_i^T \mathbf{e}_j)$ and the hadamard product (HMLP) $f_{\text{HMLP}}(v_i, v_j) = f_{\text{MLP}}(\mathbf{e}_i \circ \mathbf{e}_j)$. Both of these decoders are symmetric, (*i.e.* $f(v_i, v_j) = f(v_j, v_i)$), which is a desirable property in undirected graphs but might not be so desirable in undirected settings. It is straight forward to extend both DP and HMLP to directed settings through the insertion of a learnable matrix that looks spiritually like a learnable metric tensor. Doing so, we introduce two variants we term matrix dot product (mDP), $f_{\text{mDP}}(v_i, v_j) = \sigma(\mathbf{e}_i^T \mathbf{W} \mathbf{e}_j)$, and matrix HMLP, $f_{\text{mHMLP}}(v_i, v_j) = f_{\text{MLP}}(\mathbf{W}\mathbf{e}_i \circ \mathbf{e}_j)$, where $\mathbf{W}$ is a learnable matrix. In addition, we define a trivially asymmetric decoder named Concat MLP (CMLP) and its matrix extension, matrix Concat MLP (mCMLP), defined as:

$$f_{\text{CMLP}} = f_{\text{MLP}}(\mathbf{e}_i \,\big\|\, \mathbf{e}_j) \text{ and } f_{\text{mCMLP}} = f_{\text{MLP}}(\mathbf{W}\mathbf{e}_i \,\big\|\, \mathbf{e}_j). \tag{8}$$

## 4 ANALYSIS OF DIRECTIONALITY FOR LINK PREDICTION

In this section, we consider each of the design principles independently and perform experiments that allow us to understand the impact of directionality on each one. All experiments are performed on three different datasets: CORA, CHAMELEON, and BLOG, The results are averaged over ten runs, and and the relevant hyperparameters are optimized using OPTUNA (Akiba et al., 2019). [1]

**Directed vs Undirected Graph Encoders.** Traditionally, the form and structure of the graph encoder have been the main area of focus in the GNN literature. Thus, we explore multiple GNN encoding architectures to understand the extent to which directionally aware graph encoders impact

---

[1] In the ablation study results presented in Tables 1-5, the best-performing version for each dataset is highlighted in orange.

the predictive performance. We constructed an experiment where comparing two standard graph encoders - a GCN and GraphSAGE – with a directionally aware convolution, DirGNN. We performed this comparison by holding the decoder fixed and conducting a hyperparameter search over the relevant hyperparameters of the encoder itself. We present the results of the experiment in Table 1.

Table 1: Encoder comparisons in terms of MRR.

| Encoder | CORA | CHAMELEON | BLOG |
|---|---|---|---|
| GCN | $0.401_{\pm 0.062}$ | $0.595_{\pm 0.058}$ | $0.283_{\pm 0.039}$ |
| GraphSage | $0.414_{\pm 0.077}$ | $0.603_{\pm 0.062}$ | $0.275_{\pm 0.027}$ |
| DirGNN | $0.503_{\pm 0.088}$ | $0.609_{\pm 0.026}$ | $0.278_{\pm 0.024}$ |

We observe that in two of three datasets, DirGNN performs better than the other encoders. It is interesting to note that the "uplift offered by more complex encoders provides only modest gains. This suggests that in directed settings, DirGNN is a sensible first choice as a graph encoder because it either performs better than others, or is within the error bars of the best.

**Directed vs Undirected Graph Decoders.** Because link prediction is traditionally viewed as an edge-wise prediction task with a link predictor taking the form $f(v_i, v_j) \to \mathbb{R}^+$, we next explore whether learning a link predictor with an asymmetric decoder (*e.g.*, $f(v_i, v_j) \neq f(v_j, v_i)$) leads to better predictive performance.

Table 2: Decoder comparisons in terms of MRR.

| Decoder | CORA | CHAMELEON | BLOG |
|---|---|---|---|
| DP | $X_{\pm X}$ | $0.303_{\pm 0.021}$ | $0.115_{\pm 0.016}$ |
| HMLP | $0.178_{\pm 0.046}$ | $0.261_{\pm 0.047}$ | $0.113_{\pm 0.024}$ |
| CMLP | $0.500_{\pm 0.087}$ | $0.289_{\pm 0.102}$ | $0.160_{\pm 0.023}$ |
| mDP | $0.247_{\pm 0.066}$ | $0.214_{\pm 0.095}$ | $0.105_{\pm 0.028}$ |
| mHMLP | $0.621_{\pm 0.489}$ | $0.320_{\pm 0.066}$ | $0.147_{\pm 0.055}$ |
| mCMLP | $0.248_{\pm 0.072}$ | $0.136_{\pm 0.082}$ | $0.131_{\pm 0.023}$ |

To explore this, we constructed an experiment where the encoder was held fixed, and varied the decoder over two symmetric and four asymmetric decoders. For mathematical definitions of all decoders, please see Section 3.2. The results of this experiment are reported in Table 2. We observe that asymmetric decoders outperform symmetric ones across all three datasets, confirming our intuition that asymmetry is an important property to capture. Within the asymmetric decoders, we find that both mHMLP and CMLP outperform the others. Because mHMLP amounts to learning a pseudo-metric which can be unstable due to the many possible degeneracies, we use CMLP in the subsequent work.

**Directed vs Undirected Labeling Tricks.** Labeling tricks are one technique for breaking the node-automorphism symmetry which limits the expressivity of GNNs for link prediction Zhang et al. (2021a). To do this, the node-features of a vertex are augmented by labels that connote some structural information. Popular labeling tricks include distance encoding (Li et al., 2020) and double radius node labeling (Zhang & Chen, 2018). However, in this work, we consider only the directed extension of distance encoding due to its simplicity. Indeed, the directed extension of the distance encoding described in (Li et al., 2020) simply involves computing the distance between two vertices in a directed fashion as defined in Equation 9. It is canonical to define a maximum distance to limit the computational expense of path finding. In undirected settings, this maximum distance is often on the order of 3-5 (Chamberlain et al., 2023). In directed settings, this maximum may need to be larger to account for the fact that $d_{dir}(u, v) \geq d_{undir}(u, v)$.

Table 3: Comparison of undirected and directed labeling tricks in terms of MRR.

| Labeling | CORA | CHAMELEON | BLOG |
|---|---|---|---|
| de3-u | $0.283_{\pm 0.044}$ | $0.398_{\pm 0.069}$ | $0.137_{\pm 0.024}$ |
| de15-u | $0.324_{\pm 0.049}$ | $0.385_{\pm 0.109}$ | $0.116_{\pm 0.031}$ |
| delog-u | $0.270_{\pm 0.049}$ | $0.392_{\pm 0.116}$ | $0.120_{\pm 0.017}$ |
| de3-d | $0.498_{\pm 0.071}$ | $0.289_{\pm 0.102}$ | $0.160_{\pm 0.023}$ |
| de15-d | $0.493_{\pm 0.067}$ | $0.405_{\pm 0.065}$ | $0.125_{\pm 0.035}$ |
| delog-d | $0.305_{\pm 0.050}$ | $0.397_{\pm 0.077}$ | $0.109_{\pm 0.027}$ |

To understand the impact of a directed distance encoding on the predictive performance of GNN, we conducted an experiment where all modeling parameters were held fixed, and only the labeling trick was varied. The results are reported in Table 3. For the labeling trick, we constructed three variants of both the directed and undirected distance encodings. These variants are de3, de5, delog; which correspond to distance encodings with a maximum distance of 3, 15, and no maximum but log-transformed.

The -d and -u labels indicate that the method is directed or undirected, respectively. In these experiments, we observe that the directionality provides improvements across all datasets, but the size of impact varies significantly. In the example of CORA, we observe a 50% improvement, while both CHAMELEON and BLOG have much more modest gains. Interestingly, we find that the maximum distance cutoff does not correlate in a predictable fashion with performance. Based on these results,

we conclude that directionality should be accounted for when using a labeling trick during modeling, and that the maximum distance cutoff should be carefully tuned.

**Directed vs Undirected Negative Sampling.** We next turn our attention to exploring the effects of directionality in negative sampling. Link prediction is traditionally constructed as a binary classification task, where positive samples are observed edges and negative samples are unobserved edges. Enumerating all negative samples is intractable, so it is common to sample a subset of those negatives for training. It is possible to generate negative samples in either a directed or an undirected fashion. Directed negative sampling generates a dataset $\mathcal{D}^n = \{(u,v) : u,v \sim \mathcal{V}\}$ with $(u,v) \neq (v,u)$. In undirected negative sampling, $\mathcal{D}^n = \{(u,v) : u,v \sim \mathcal{V}\}$ with $(u,v) = (v,u)$.

Table 4: Comparison of undirected and directed negative sampling in terms of MRR.

| Sampling | CORA | CHAMELEON | BLOG |
|---|---|---|---|
| Undirected | $0.48_{\pm 0.12}$ | $0.60_{\pm 0.07}$ | $0.27_{\pm 0.03}$ |
| Directed | $0.50_{\pm 0.09}$ | $0.61_{\pm 0.03}$ | $0.28_{\pm 0.02}$ |

To explore this design principle, we constructed an experiment where we held the model parameters fixed, and altered only the negative sampling strategy. In this experiment we used DirGNN as the encoder, CMLP as the decoder, de15 as the labeling method, and all structural features. The results for this experiment are presented in Table 4. We note across all three datasets that directed negative sampling provides a lift in MRR, although in two of the three datasets, the lift is modest. We conclude that there is evidence to support the usage of directed negative sampling, but that these effects are smaller than other design principles.

**Directed vs Undirected Structural Features.** Previous work has shown that the inclusion of edge-wise structural features, such as the number of shared common neighbors at k-hops, leads to significant performance improvements for link prediction (Zhang et al., 2024; Ai et al., 2022; Zhang & Chen, 2018). Indeed, this intuitively makes sense because these structural features are the building blocks for heuristic similarity measures such as Adamic-Adar or Resource-Allocation, both of which represent strong baselines in undirected link prediction settings. The definition of our structural features can be found in Equations 13-15.

Table 5: Comparison of undirected and directed structural features in terms of MRR.

| SFs | CORA | CHAMELEON | BLOG |
|---|---|---|---|
| undirected | $0.309_{\pm 0.051}$ | $0.348_{\pm 0.020}$ | $0.174_{\pm 0.020}$ |
| directed | $0.412_{\pm 0.064}$ | $0.534_{\pm 0.038}$ | $0.251_{\pm 0.033}$ |

To understand whether directionality affects the performance of models that include structure features, we constructed an experiment where we used DirGNN as the encoder, and CMLP as the decoder, while varying the structural features. The results are reported in Table 5. We observe that across all three datasets, directed structural features provide a significant improvement over their undirected variants. In percentage-increase terms, we find an uplift of 30% to 50% through the inclusion of directionality, indicating that this is a significant design principle. The improvement is larger than the associated uncertainties, giving us confidence that this improvement is robust.

## 5 PROPOSED MODEL

Based on the insights drawn through the experiments conducted on analyzing directionality, we propose a framework, namely *DirLP*, for directed link prediction. DirLP features the following key components: A directed labeling trick, a directed encoder, a directed structure feature extractor, and an asymmetric decoder. An overview of model architecture is provided in Figure 5. In the remainder of this section, we describe each element and how they're combined to form DirLP.

We begin with the **directed labeling trick**, which injects structural information into our graph encoder, and is defined as:

$$\mathbf{l}_t = \left[ \mathrm{d}^\delta(t,v) \,\middle\|\, \mathrm{d}^\delta(t,v) \forall v \in \mathcal{V} \right] \tag{9}$$

where $\mathrm{d}^\delta$ is the truncated graph distance defined as $\mathrm{d}^\delta(t,v) = \min(\delta, d(t,v))$, $t$ is the landmark vertex, and $\delta$ is the maximum distance. For DirLP, we use two fixed landmarks.

We utilize **Directed Graph Neural Network (DirGNN) as encoder**, which aggregates messages from incoming and outgoing edges separately for each node, and obtain layer updates by a non-parametric combinator function (Rossi et al., 2024). More formally, DirGNN initializes the hidden node embeddings by intermediate node features, i.e., $\mathbf{h}_u^{(0)} = \mathbf{x}_u'$ for all $u \in \mathcal{V}$. With our choices of

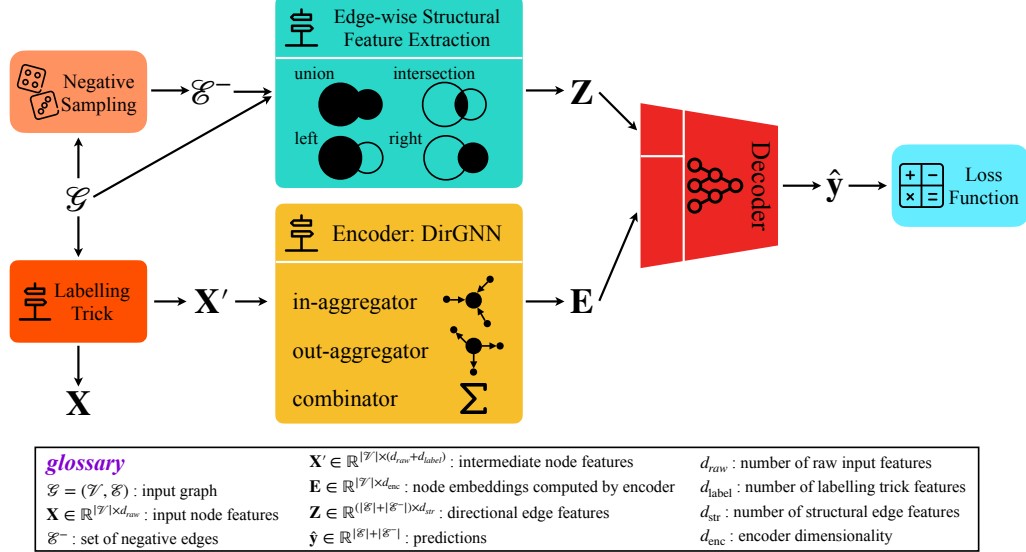

Figure 2: **Overview of DirLP.** Given the input graph $\mathcal{G} = (\mathcal{V}, \mathcal{E})$ and node features $\mathbf{x} \in \mathbb{R}^{d_{\mathrm{raw}}}, \forall u \in \mathcal{V}$, DirLP follows a series of steps to predict directed links. First, a set of negative edges $\mathcal{E}^-$ is generated. Next, for each edge $(u, v)$ in the set $\mathcal{E} \cup \mathcal{E}^-$, structural edge features $\mathbf{z}_{(u,v)} \in \mathbb{R}^{d_{\mathrm{str}}}$ are computed. Then, directional labels are assigned to each node $u \in \mathcal{V}$, and intermediate node features $\mathbf{x}'_u \in \mathbb{R}^{d_{\mathrm{raw}} + d_{\mathrm{label}}}$ are constructed by concatenating the original node features with the directional labels. The model then applies DirGNN message passing to produce node embeddings $\mathbf{e}_u \in \mathbb{R}^{d_{\mathrm{enc}}}$ for $u \in \mathcal{V}$. For each edge $(u, v)$ in $\mathcal{E} \cup \mathcal{E}^-$, the edge features are concatenated with the node embeddings of the edge's endpoints. Finally, these concatenated embeddings are passed through an MLP followed by a sigmoid activation function to make predictions.

aggregation and update functions, GraphSage (Hamilton et al., 2017) and convex combination (Rossi et al., 2024), respectively, at the $k^{\mathrm{th}}$ layer of encoder node embeddings $\mathbf{h}_u^{(k)}$ are updated as follows:

$$\mathbf{m}_{u,\mathrm{in}}^{(k+1)} = \mathbf{W}_{\mathrm{in,self}}^{(k)} \mathbf{h}_u^{(k)} + \mathbf{W}_{\mathrm{in}}^{(k)} \frac{\sum_{v \in \mathcal{N}_{\mathrm{in}}(u)} \mathbf{h}_v^{(k)}}{|\mathcal{N}_{\mathrm{in}}(u)|} \tag{10}$$

$$\mathbf{m}_{u,\mathrm{out}}^{(k+1)} = \mathbf{W}_{\mathrm{out,self}}^{(k)} \mathbf{h}_u^{(k)} + \mathbf{W}_{\mathrm{out}}^{(k)} \frac{\sum_{v \in \mathcal{N}_{\mathrm{out}}(u)} \mathbf{h}_v^{(k)}}{|\mathcal{N}_{\mathrm{out}}(u)|} \tag{11}$$

$$\mathbf{h}_u^{(k+1)} = \alpha \times \mathbf{m}_{u,\mathrm{in}}^{(k+1)} + (1 - \alpha) \times \mathbf{m}_{u,\mathrm{out}}^{(k+1)}, \tag{12}$$

where $\mathbf{W}_{\mathrm{in}}^{(k)}, \mathbf{W}_{\mathrm{in,self}}^{(k)}, \mathbf{W}_{\mathrm{out}}^{(k)}, \mathbf{W}_{\mathrm{out,self}}^{(k)}$ are learnable parameters and $\alpha$ is a hyperparameter that controls the trade-off of emphasis between incoming and outgoing edges. In our experiments, we set $\alpha = 0.5$ for all datasets to equally treat directions. The node embeddings are set to final layer output, i.e., $\mathbf{e}_u = \mathbf{h}_u^{(K)}, \forall u \in \mathcal{V}$, where $K$ denotes the total number of update layers.

We perform **edge-wise structural feature extraction** to incorporate the directionally aware structural information at the edge-level. We define a set of *neighbourhood directionality sequences* at length $n$, $\mathbb{S}_n = \{(s_1, \ldots, s_n) : \forall s_i \in \{\mathrm{in}, \mathrm{out}\}\}$. Now, for a given node $u$ and directionality sequence $\mathbf{s} = (s_1, \ldots, s_n)$, we define directional neighbourhood $\mathcal{N}_{\mathbf{s}}^{\mathrm{dir}}(u)$ such that $v \in \mathcal{N}_{\mathbf{s}}^{\mathrm{dir}}(u)$, if and only if $v$ is reachable from $u$ with an $n$-step walk where $i^{\mathrm{th}}$ step is in the direction of $s_i$. For an edge $(u, v)$, we compute the cardinality of the union (U) and intersection (I) of the directed neighborhoods of endpoints, in addition to the individual neighbourhoods of left (L) and right (R) side as

follows:

$$\mathbf{z}^{\mathrm{U}}_{(u,v)} = [|\mathcal{N}_{\mathbf{s}_1}(u) \cup \mathcal{N}_{\mathbf{s}_2}(v)|]_{\mathbf{s}_1, \mathbf{s}_2 \in \bigcup_{n=1}^{N} \mathbb{S}_n}, \quad \mathbf{z}^{\mathrm{L}}_{(u,v)} = [|\mathcal{N}_{\mathbf{s}}(u)|]_{\mathbf{s} \in \bigcup_{n=1}^{N} \mathbb{S}_n}, \quad (13)$$

$$\mathbf{z}^{\mathrm{I}}_{(u,v)} = [|\mathcal{N}_{\mathbf{s}_1}(u) \cap \mathcal{N}_{\mathbf{s}_2}(v)|]_{\mathbf{s}_1, \mathbf{s}_2 \in \bigcup_{n=1}^{N} \mathbb{S}_n}, \quad \mathbf{z}^{\mathrm{R}}_{(u,v)} = [|\mathcal{N}_{\mathbf{s}}(v)|]_{\mathbf{s} \in \bigcup_{n=1}^{N} \mathbb{S}_n}, \quad (14)$$

$$\mathbf{z}^{\mathrm{dir}}_{(u,v)} = \mathbf{z}^{\mathrm{U}}_{(u,v)} \,\big\|\, \mathbf{z}^{\mathrm{I}}_{(u,v)} \,\big\|\, \mathbf{z}^{\mathrm{L}}_{(u,v)} \,\big\|\, \mathbf{z}^{\mathrm{R}}_{(u,v)}, \quad (15)$$

where $N$ is the maximum radius. Next, we compute the undirected versions over symmetrized adjacency matrix that define neighbourhood $\mathcal{N}_k(u)$ which involves nodes at $k$-hop distance to a given node $u$:

$$\mathbf{z}^{\mathrm{undir}}_{(u,v)} = [|\mathcal{N}_k(u) \cup \mathcal{N}_k(v)|]_{k=1}^{N} \,\big\|\, [|\mathcal{N}_k(u) \cap \mathcal{N}_k(v)|]_{k=1}^{N} \,\big\|\, [|\mathcal{N}_k(u)|]_{k=1}^{N} \,\big\|\, [|\mathcal{N}_k(v)|]_{k=1}^{N}. \quad (16)$$

Finally, the complete edge feature vector is obtained by concatenating the directed and undirected structural features; $\mathbf{z}_{(u,v)} = \mathbf{z}^{\mathrm{dir}}_{(u,v)} \,\|\, \mathbf{z}^{\mathrm{undir}}_{(u,v)}$.

We employ a simple **feedforward network as decoder** that concatenates the edge features and node embeddings of source and target node and performs linear transformations through a *Multi-layer Perceptron (MLP)*:

$$\hat{y}_{u,v} = \sigma \left( f_{\mathrm{MLP}} \left( \mathbf{z}_{u,v} \,\big\|\, \mathbf{e}_u \,\big\|\, \mathbf{e}_v \right) \right), \quad (17)$$

where $f_{\mathrm{MLP}}(\cdot)$ is a feed forward network and $\sigma$ is the sigmoid function.

**Expressivity.** Being able to represent edge direction through an asymmetric decoder and including information about directed triangle counts, DirLP is capable of distinguishing edges that conventional MPNNs are not able to which is stated by Theorem 5.1.

**Theorem 5.1** *Let $\mathcal{M}_{\mathrm{sGNN}}$ be the family of GNNs defined by Equation 7 equipped with a symmetric decoder and augmented by undirected structural features. Additionally, let $\mathcal{M}_{\mathrm{DirLP}}$ be family of all models defined by Equation 17. $\mathcal{M}_{\mathrm{DirLP}}$ is strictly more powerful than $\mathcal{M}_{\mathrm{sGNN}}$ ($\mathcal{M}_{\mathrm{sGNN}} \subset \mathcal{M}_{\mathrm{DirLP}}$).*

This makes intuitive sense because our asymmetric decoder can represent all symmetric decoders, which allows DirLP to distinguish all links any sGNN can. We present the proof in Appendix C.

## 6 PRINCIPLE COMPARISON

Table 6: Comparison of baseline methods and our proposed model in terms MRR on directed link prediction task. The top three models are highlighted as **First**, **Second**, **Third**. Note that The Blog dataset does not have vertex features and therefore MLP model built from vertex features do not apply.

| | CORA | CITESEER | CHAMELEON | SQUIRREL | BLOG | WIKICS |
|---|---|---|---|---|---|---|
| LP, sym | $0.315_{\pm 0.065}$ | $0.303_{\pm 0.073}$ | $0.235_{\pm 0.013}$ | $0.102_{\pm 0.005}$ | $0.096_{\pm 0.021}$ | $0.661_{\pm 0.011}$ |
| LP, asym | $0.324_{\pm 0.056}$ | $0.146_{\pm 0.024}$ | $0.381_{\pm 0.075}$ | $0.497_{\pm 0.151}$ | $0.149_{\pm 0.030}$ | $0.424_{\pm 0.067}$ |
| RA, sym | $0.356_{\pm 0.07}$ | $0.293_{\pm 0.049}$ | $0.194_{\pm 0.069}$ | $0.087_{\pm 0.013}$ | $0.082_{\pm 0.021}$ | $0.358_{\pm 0.074}$ |
| RA, asym | $0.292_{\pm 0.046}$ | $0.134_{\pm 0.021}$ | $0.353_{\pm 0.142}$ | $0.148_{\pm 0.037}$ | $0.103_{\pm 0.024}$ | $0.494_{\pm 0.101}$ |
| AA, sym | $0.353_{\pm 0.067}$ | $0.254_{\pm 0.048}$ | $0.239_{\pm 0.016}$ | $0.103_{\pm 0.005}$ | $0.096_{\pm 0.023}$ | $0.285_{\pm 0.033}$ |
| AA, asym | $0.288_{\pm 0.045}$ | $0.122_{\pm 0.02}$ | $0.378_{\pm 0.091}$ | $0.495_{\pm 0.196}$ | $0.143_{\pm 0.034}$ | $0.487_{\pm 0.060}$ |
| MLP | $0.172_{\pm 0.03}$ | $0.356_{\pm 0.084}$ | $0.104_{\pm 0.029}$ | $0.051_{\pm 0.027}$ | - | $0.019_{\pm 0.006}$ |
| GAT | $0.087_{\pm 0.036}$ | $0.115_{\pm 0.043}$ | $0.150_{\pm 0.026}$ | $0.088_{\pm 0.038}$ | $0.057_{\pm 0.011}$ | $0.094_{\pm 0.010}$ |
| GCN | $0.402_{\pm 0.062}$ | $0.208_{\pm 0.048}$ | $0.270_{\pm 0.043}$ | $0.260_{\pm 0.018}$ | $0.080_{\pm 0.025}$ | $0.277_{\pm 0.047}$ |
| GraphSage | $0.414_{\pm 0.077}$ | $0.158_{\pm 0.063}$ | $0.202_{\pm 0.046}$ | $0.190_{\pm 0.074}$ | $0.083_{\pm 0.016}$ | $0.185_{\pm 0.058}$ |
| DirLP | $0.504_{\pm 0.088}$ | $0.480_{\pm 0.108}$ | $0.657_{\pm 0.037}$ | $0.759_{\pm 0.012}$ | $0.280_{\pm 0.031}$ | $0.752_{\pm 0.028}$ |

**Setup.** In our experiments, we generated ten sets of random splits of datasets for training, validation, and testing to facilitate 10-fold cross-validation. These dataset splits will be made publicly available upon official publication of this work. Each of the deep-learning models was optimized using the hyperparameter tuning framework OPTUNA (Akiba et al., 2019), with 48 optimization

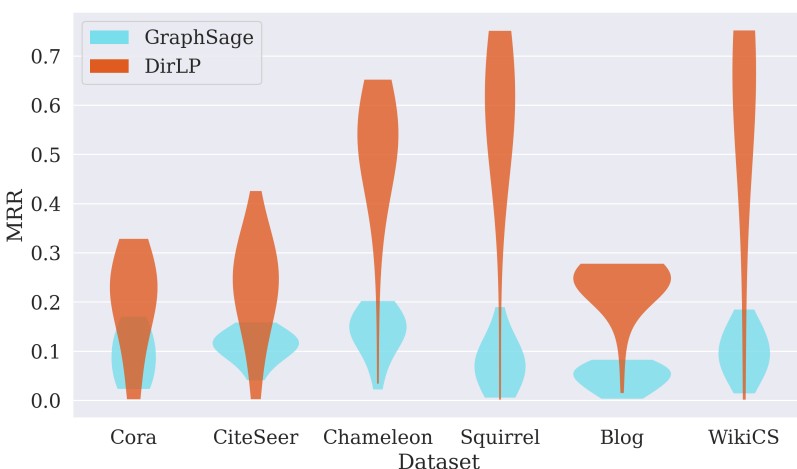

Figure 3: **Statistical Comparison Between GraphSage and DirLP.** The violin plots illustrate the performance of GraphSage and DirLP in terms of MRR across multiple data splits.

steps performed per model to find the best-performing configurations. The optimization process was conducted on the validation set focusing on the Mean Reciprocal Rank (MRR). The search space of OPTUNA and tuned hyperparameter settings for all deep-learning models are provided in Appendix E. Experiments were run on an NVIDIA DGX A100 machine with 128 AMD ROME 7742 cores and 8 NVIDIA A100 GPUs, utilizing PYTORCH GEOMETRIC 2.5.3 and PYTORCH 2.3.1 for model training and evaluation.

**Baselines.** We performed a set of principle comparison experiments between our proposed method, DirLP, and several baseline approaches, including both symmetric and asymmetric versions of similarity-based heuristics LP (Lü et al., 2009), RA (Zhou et al., 2009), and AA (Adamic & Adar, 2003), as well as four deep-learning based baselines: MLP, GAT (Veličković et al., 2017), GCN (Kipf & Welling, 2016) and GraphSage (Hamilton et al., 2017). The implementations of deep-learning models were based on official code provided in PYTORCH GEOMETRIC 2.5.3, ensuring consistency and reproducibility.

**Results.** The main results of our principle comparison experiments are presented in Table 6 in terms of MRR. Additionally, in Appendix D, we report the performance comparison in terms of Hits@20 in the Appendix in Table 8. Based on these results, we draw several important conclusions. First, we observe that the asymmetric versions of heuristic methods consistently outperform their symmetric counterparts on datasets such as CHAMELEON, SQUIRREL, BLOG, and WIKICS, with the only exception of LP on WIKICS. Reviewing the dataset statistics reported in Appendix A, we find a positive correlation between graph density and the performance advantage of using asymmetric methods. This suggests that as the complexity of the graph structure increases, the inclusion of directional information becomes more critical.

In all cases, heuristic methods outperform deep-learning baselines that do not incorporate directionality in the message-passing framework. This finding suggests that incorporating edge directionality can have a greater impact than increasing model complexity, particularly in many settings. Deep-learning baselines perform poorly, especially in cases where the node features offer limited information. For instance, on datasets BLOG and WIKICS, which have relatively low vertex feature dimensionality compared to other datasets (see Appendix A), heuristic methods significantly outperform the deep-learning baselines. This highlights the importance of effectively modeling directionality in graph structure when node features are insufficient.

The principle comparison experiments clearly demonstrate the superiority of DirLP, which captures directionality through message-passing mechanisms and feature extraction both at the edge and node level. In many instances, DirLP delivers significantly superior performance compared to deep-learning baselines, highlighting its ability to model directional relationships more effectively.

**Sensitivity.** A violin plot [2] illustrating the comparison between DirLP and GraphSage, in terms of MRR, is shown in Figure 6, offering further insights into the model's performance distribution across the data splits. It is observed that DirLP shows higher variance, however on the average it outperforms GraphSage clearly.

# 7 CONCLUSION

In conclusion, this paper introduces DirLP, a novel framework for directed link prediction that outperforms existing models across benchmark datasets. By leveraging an asymmetric decoder and directed structural features, DirLP effectively captures relationships in graphs where edge directionality is critical, highlighting the limitations of traditional undirected methods.

Rather than introducing a complex new architecture, our work focuses on systematically exploring the utility of simple, directed variants of existing techniques. Directed distance encoding and directed GNNs, though seemingly minor modifications, demonstrate substantial performance gains, emphasizing the practical value of incorporating directionality. This study serves as a guide for practitioners, showing how fundamental, interpretable methods can deliver strong results in directed settings.

While scalability remains a consideration due to preprocessing costs of edge-wise structural feature extraction, these are one-time operations that can be optimized. Future work will focus on enhancing the efficiency of our approach and extending evaluations to larger datasets to broaden its applicability.

This work sets a benchmark for directed link prediction and lays a foundation for future research, encouraging deeper exploration into the role of directionality and the development of scalable, high-performing solutions for directed graph tasks.

---

[2]Violin plots provides a visual summary of the data distribution along with its probability density that is smoothed and symmetrized by a Kernel density estimation (KDE).

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

## A  DATASET DETAILS

In our experiments, we evaluate directed link prediction performance of various approaches using six benchmark datasets: CORA (Yang et al., 2016), CITESEER (Yang et al., 2016), CHAMELEON (Rozemberczki et al., 2021), SQUIRREL (Rozemberczki et al., 2021), BLOG (He et al., 2022), WIKICS (Mernyei & Cangea, 2020). CORA and CITESEER are citation networks where the nodes denote the papers and links denote the citations from one to another. Likewise, CHAMELEON, SQUIRREL and WIKICS are reference networks on Wikipedia pages in the corresponding topics where edges reflect reference links between them. BLOG is a set of political blogs from the 2004 US presidential election with links recording mentions between them.

Table 7: Dataset Statistics. $d$, $|\mathcal{V}|$, $|\mathcal{E}|$, $|\mathcal{E}_{\leftrightarrow}|/|\mathcal{E}|$, denote the number of node features, number of nodes, number of edges and ratio of node pairs connected in both direction to total number of edges, respectively. Grap density is calculated by $|\mathcal{E}|/(|\mathcal{V}| \times (|\mathcal{V}| - 1))$.

|  | CORA | CITESEER | CHAMELEON | SQUIRREL | BLOG | WIKICS |
|---|---|---|---|---|---|---|
| $d$ | $1,433$ | $3,703$ | $2,325$ | $2,089$ | $0$ | $300$ |
| $|\mathcal{V}|$ | $2,708$ | $3,327$ | $2,277$ | $5,201$ | $1,222$ | $11,701$ |
| $|\mathcal{E}|$ | $10,556$ | $9,104$ | $36,101$ | $217,073$ | $19,024$ | $297,110$ |
| $|\mathcal{E}_{\leftrightarrow}|/|\mathcal{E}|$ | $6.1\%$ | $4.9\%$ | $26\%$ | $17\%$ | $24\%$ | $52\%$ |
| Density | $0.1\%$ | $0.1\%$ | $0.7\%$ | $0.8\%$ | $1.3\%$ | $0.2\%$ |

## B  HEURISTICS FORMULATIONS

In our experiments we use three different node similarity scores and their directed variants; local path index (LP), resource allocation index (RA), Adamic–Adar index (AA). Their original formulations (symmetric versions) are formulated as follows for a pair of nodes $u, v \in \mathcal{V}$:

$$S_{\text{LP,sym}}(u, v) = \hat{\mathbf{A}}_{u,v}^2 + \epsilon \hat{\mathbf{A}}_{u,v}^3, \tag{18}$$

$$S_{\text{RA,sym}}(u, v) = \sum_{t \in \mathcal{N}(u) \cap \mathcal{N}(v)} \frac{1}{|\mathcal{N}(t)|}, \tag{19}$$

$$S_{\text{AA,sym}}(u, v) = \sum_{t \in \mathcal{N}(u) \cap \mathcal{N}(v)} \frac{1}{\log |\mathcal{N}(t)|}. \tag{20}$$

where $\hat{\mathbf{A}}$ denotes the symmetrized adjacency matrix and $\epsilon$ is a a free parameter set to $10^{-3}$ in our experiments. Asymmetric variant of LP is simply defined as follows:

$$S_{\text{LP,asym}}(u, v) = \mathbf{A}_{u,v}^2 + \epsilon \mathbf{A}_{u,v}^3. \tag{21}$$

Recall the definition for directed neighborhood operator:

$$\mathcal{N}_{\text{in}}(u) = \{v \in \mathcal{V} \mid (v, u) \in \mathcal{E}\}, \ \mathcal{N}_{\text{out}}(u) = \{v \in \mathcal{V} \mid (u, v) \in \mathcal{E}\}, \tag{22}$$

where $\mathcal{N}_{\text{in}}(u)(\mathcal{N}^{\text{out}}(u))$ consists of all nodes that have a directed edge pointing toward (originating from) node $u$. The four directional variants of AA and RA for a given node pair $u$ and $v$ follows:

$$S_{\text{RA}, d_u - d_v}(u, v) = \sum_{t \in \mathcal{N}_{d_u}(u) \cap \mathcal{N}_{d_v}(v)} \frac{1}{|\mathcal{N}(t)|}, \tag{23}$$

$$S_{\text{AA}, d_u - d_v}(u, v) = \sum_{t \in \mathcal{N}_{d_u}(u) \cap \mathcal{N}_{d_v}(v)} \frac{1}{\log |\mathcal{N}(t)|}. \tag{24}$$

The asymmetric variants of AA and RA on our baseline experiments are selected based on best performing version of directed common neighbourhood operator.

## C  PROOF OF THEOREM 5.1

Our proof follows two steps. First, we show that $\mathcal{M}_{\text{sGNN}} \subseteq \mathcal{M}_{\text{DirLP}}$. Next, Then, we construct a graph that exhibits an automorphic nodal structure for all elements of $\mathcal{M}_{\text{sGNN}}$, but not for any element of $\mathcal{M}_{\text{DirLP}}$.

The first half of the proof can be seen by examining the structure of DirLP, and the way in which it generalizes a GNN with a symmetric decoder and symmetric structural features. The general form for a sGNN is given by:

$$f_{\text{sGNN}} = f_{\text{mlp}}(\mathbf{z}_{u,v}^{\text{undir}} || \mathbf{e}_u \circ \mathbf{e}_v). \tag{25}$$

Starting from Equation 17, we see immediately that we can recover a symmetric for decoder for $\mathbf{e}_u \| \mathbf{e}_v$ by selecting an initial layer that corresponds to two concatenated identity matrices. Turning our attention to the structural features, we again see that a special combination of the elements of

$\mathbf{z}_{u,v}$ allows us to recover $\mathbf{z}_{u,v}^{\mathrm{undir}}$. Namely, a trace over all modes allows us to construct the undirected structural features. This amounts to using a matrix of all 1s for the initial layer of our MLP. Putting this all together, we observe that we can convert our set of directed input features to their undirected variants using an MLP whose initial weight matrix is equal to $[\mathbf{B} \,\|\, \mathbf{I}_{D \times D} \,\|\, \mathbf{I}_{D \times D}]$, where $\mathbf{B}$ is the desired trace matrix. Therefore, because there exists a DirLP that is equivalent to a sGNN, any edge that is distinguished by an sGNN must also be distinguishable by a DirLP. This is sufficient to prove that $\mathcal{M}_{\mathrm{sGNN}} \subseteq \mathcal{M}_{\mathrm{DirLP}}$.

We next turn to the task of making this relationship strict, which we do by constructing a graph for which sGNN cannot distinguish a node pair but DirLP can. To do this, we consider a complete graph with four nodes as shown in Figure 4 and consider the edges $(v_0, v_1)$ and $v_0, v_3)$. In the undirected setting, all vertices exist in the same orbit, which means that no MPNN will be able to distinguish these two edges (Srinivasan & Ribeiro, 2019). Looking further, both edges have the same structural features. As a result, $\mathrm{f}_{\mathrm{sGNN}}(v_0, v_1) = \mathrm{f}_{\mathrm{sGNN}}(v_0, v_3)$. In the directed setting, it is sufficient to show that $\mathbf{z}_{u,v}$ provides different representations for these two edges. Indeed, we observe that $\mathbf{z}_{0,1} = [\mathcal{N}_0^{in}, \mathcal{N}_0^{out}, \mathcal{N}_1^{in}, \mathcal{N}_1^{out}...] = [2, 1, 1, 2, ...]$, while $\mathbf{z}_{0,3} = [2, 1, 3, 1...]$, where we have neglected the intersection and union features for convenience. We conclude that these two representations are indeed different, which completes our proof.

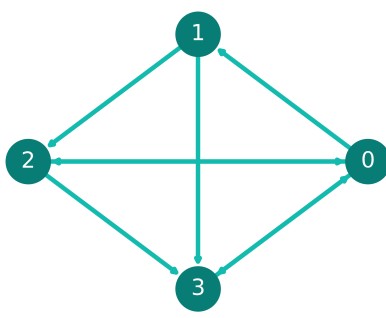

Figure 4: A complete graph with four nodes.

## D PRINCIPLE COMPARISON IN TERMS OF HITS@20

Another popular performance metric used in link prediction is Hits@$k$ which measures the proportion of correct links (positive samples) ranked within the top $k$ positions of a sorted list and formulated as follows:

$$\mathrm{Hits@}k = \frac{1}{|\mathcal{E}_{\mathrm{test}}|} \sum_{(u,v) \in \mathcal{E}_{\mathrm{test}}} \mathbb{I}\left(\mathrm{rank}(u, v) \leq k\right), \tag{26}$$

where $\mathbb{I}$ is the indicator function that returns 1 if the condition inside is true and 0 otherwise.

Table 8: The Hits@20 for models with top MRRs in Table 6. The top three models are highlighted as **First**, **Second**, **Third**.

|  | CORA | CITESEER | CHAMELEON | SQUIRREL | BLOG | WIKICS |
|---|---|---|---|---|---|---|
| LP, sym | $0.555_{\pm 0.042}$ | $0.500_{\pm 0.023}$ | $0.408_{\pm 0.006}$ | $0.168_{\pm 0.008}$ | $0.262_{\pm 0.025}$ | $0.484_{\pm 0.045}$ |
| LP, asym | $0.378_{\pm 0.014}$ | $0.151_{\pm 0.013}$ | $0.562_{\pm 0.033}$ | $0.702_{\pm 0.011}$ | $0.369_{\pm 0.027}$ | $0.629_{\pm 0.016}$ |
| RA, sym | $0.591_{\pm 0.011}$ | $0.328_{\pm 0.016}$ | $0.434_{\pm 0.035}$ | $0.175_{\pm 0.003}$ | $0.250_{\pm 0.026}$ | $0.571_{\pm 0.032}$ |
| RA, asym | $0.320_{\pm 0.012}$ | $0.137_{\pm 0.013}$ | $0.639_{\pm 0.034}$ | $0.769_{\pm 0.014}$ | $0.336_{\pm 0.039}$ | $0.723_{\pm 0.015}$ |
| AA, sym | $0.581_{\pm 0.01}$ | $0.292_{\pm 0.017}$ | $0.432_{\pm 0.006}$ | $0.170_{\pm 0.010}$ | $0.266_{\pm 0.018}$ | $0.516_{\pm 0.042}$ |
| AA, asym | $0.315_{\pm 0.013}$ | $0.125_{\pm 0.012}$ | $0.601_{\pm 0.035}$ | $0.733_{\pm 0.009}$ | $0.379_{\pm 0.029}$ | $0.676_{\pm 0.013}$ |
| MLP | $0.343_{\pm 0.034}$ | $0.592_{\pm 0.034}$ | $0.318_{\pm 0.030}$ | $0.159_{\pm 0.066}$ | - | $0.065_{\pm 0.018}$ |
| GAT | $0.176_{\pm 0.058}$ | $0.227_{\pm 0.055}$ | $0.164_{\pm 0.031}$ | $0.030_{\pm 0.022}$ | $0.072_{\pm 0.019}$ | $0.035_{\pm 0.016}$ |
| GCN | $0.599_{\pm 0.017}$ | $0.457_{\pm 0.034}$ | $0.301_{\pm 0.044}$ | $0.103_{\pm 0.027}$ | $0.116_{\pm 0.033}$ | $0.101_{\pm 0.022}$ |
| GraphSage | $0.650_{\pm 0.012}$ | $0.416_{\pm 0.081}$ | $0.223_{\pm 0.041}$ | $0.056_{\pm 0.030}$ | $0.120_{\pm 0.020}$ | $0.051_{\pm 0.032}$ |
| DirLP | $0.767_{\pm 0.057}$ | $0.991_{\pm 0.012}$ | $0.727_{\pm 0.032}$ | $0.706_{\pm 0.013}$ | $0.384_{\pm 0.035}$ | $0.635_{\pm 0.053}$ |

# E HYPERPARAMETER SETTINGS

Table 9: Search spaces for hyperparameters.

| Hyperparameter | Sampling Distribution |
|---|---|
| # of hidden layers | {1,2,4} |
| hidden layer dim. | {32,64,128} |
| final layer dim. | {24,48,72} |
| # of heads | {2,4,8,16} |
| dropout prob. | Uniform(0, 0.9) |
| learning rate | Uniform(0.0001, 0.0600) |

Table 10: Best hyperparameter settings for GAT experiments in Table 6.

| | CORA | CITESEER | CHAMELEON | SQUIRREL | BLOG | WIKICS |
|---|---|---|---|---|---|---|
| # of hidden layers | 1 | 1 | 2 | 1 | 1 | 1 |
| hidden layer dim. | 128 | 128 | 32 | 32 | 32 | 32 |
| final layer dim. | 72 | 48 | 72 | 24 | 24 | 24 |
| # of heads | 4 | 2 | 8 | 2 | 8 | 8 |
| dropout prob. | 0.040 | 0.145 | 0.301 | 0.414 | 0.020 | 0.479 |
| learning rate | 0.010 | 0.005 | 0.043 | 0.048 | 0.059 | 0.024 |

Table 11: Best hyperparameter settings for GCN experiments in Table 6.

| | CORA | CITESEER | CHAMELEON | SQUIRREL | BLOG | WIKICS |
|---|---|---|---|---|---|---|
| # of hidden layers | 1 | 2 | 1 | 1 | 2 | 1 |
| hidden layer dim. | 64 | 128 | 64 | 64 | 64 | 32 |
| final layer dim. | 72 | 72 | 48 | 48 | 72 | 72 |
| dropout prob. | 0.003 | 0.007 | 0.331 | 0.274 | 0.144 | 0.090 |
| learning rate | 0.031 | 0.007 | 0.012 | 0.004 | 0.013 | 0.016 |

Table 12: Best hyperparameter settings for GraphSage experiments in Table 6.

| | CORA | CITESEER | CHAMELEON | SQUIRREL | BLOG | WIKICS |
|---|---|---|---|---|---|---|
| # of hidden layers | 1 | 1 | 1 | 1 | 1 | 1 |
| hidden layer dim. | 64 | 64 | 32 | 32 | 32 | 32 |
| final layer dim. | 72 | 48 | 72 | 72 | 72 | 72 |
| dropout prob. | 0.181 | 0.264 | 0.120 | 0.049 | 0.044 | 0.062 |
| learning rate | 0.017 | 0.025 | 0.005 | 0.020 | 0.032 | 0.055 |

Table 13: Best hyperparameter settings for DirLP experiments in Table 6.

| | CORA | CITESEER | CHAMELEON | SQUIRREL | BLOG | WIKICS |
|---|---|---|---|---|---|---|
| # of hidden layers | 1 | 2 | 2 | 1 | 2 | 2 |
| hidden layer dim. | 64 | 128 | 128 | 64 | 128 | 128 |
| final layer dim. | 72 | 48 | 72 | 48 | 72 | 24 |
| dropout prob. | 0.063 | 0.027 | 0.154 | 0.034 | 0.020 | 0.130 |
| learning rate | 0.045 | 0.037 | 0.037 | 0.029 | 0.090 | 0.014 |

