# OpenReview forum: "Understanding the Design Principles of Link Prediction in Directed Settings"
_ICLR.cc/2025/Conference — Submitted to ICLR 2025_

### Official Review · Reviewer_J11z · 2024-10-16

**Soundness:** 2
**Presentation:** 3
**Contribution:** 1
**Rating:** 3
**Confidence:** 4

**Summary:**

The paper addresses the challenge of directed link prediction within Graph Representation Learning (GRL), a task traditionally focused on undirected graphs. Recognizing that real-world data often contains crucial directional information, the authors highlight the limitations of using models designed for symmetric, undirected graphs in such settings. They adapt successful heuristics from undirected link prediction for directed graphs and demonstrate their effectiveness, particularly in comparison to state-of-the-art Graph Neural Networks (GNNs). Through extensive experiments, they present a new framework, DirLP, which consistently outperforms traditional and cutting-edge models in directed link prediction tasks.

**Strengths:**

1. The paper identifies and addresses a key gap in the current GRL literature by focusing on the complexities of directed interactions, which are often overlooked in favor of simpler, undirected models.
2. The authors provide a thorough comparison across heuristic, MLP, and GNN-based models for directed link prediction, offering insights into how directionality impacts predictive performance.
3. The novel DirLP framework not only adapts key design principles for directed graphs but also demonstrates superior performance over leading models on multiple benchmarks, making a significant contribution to the field.

**Weaknesses:**

1. The paper is lack of novelty. The key difference with the SEAL is the directed GNN and directed distance which is only minor modification on the original method. edge-wise structural feature extraction is a single extension of the Buddy method while without efficiency hashing method.
2. The proposed method is inefficiency. The labeling trick is with O(N^3) time complexity and O(N^2) memory complexity
3. The benchmark datasets are problematic. The utilized dataset are quite small while none of the OGB graph is included despite some are also directed.  The Chameleon and squirrel datasets without filter has some repeat edges, which may lead to data leakage[1]
4. The baseline selected is out of dated, the lastest baseline is GAT in 2018, without any GNN for link prediction baseline and directed GNN baselines.
5. The principal seems trivial that for all the components, the directed one is better than the undirected one as we focuses on the directed setting instead of undirected setting.

[1] Platonov, Oleg, et al. "A critical look at the evaluation of GNNs under heterophily: Are we really making progress?." The Eleventh International Conference on Learning Representations.

**Questions:**

1. Given the high computational complexity, how does the proposed labeling trick (O(N^3) time and O(N^2) memory complexity) impact the efficiency of the method in large-scale graphs?
2. Why were the benchmark datasets chosen relatively small, and why were OGB graphs (some of which are directed) not included in the evaluation?
3. How did the repeated edges in the Chameleon and Squirrel datasets affect the integrity of the results, particularly concerning potential data leakage?
4. Why were the baselines selected (the latest being GAT from 2018) not updated to include more recent GNNs or directed GNNs designed specifically for link prediction tasks?
5. How do the findings provide non-trivial insights, given that the result showing directed components outperforming undirected ones in directed settings might seem intuitive?

---

> ### Author Response · Authors · 2024-11-22
> **Reply**
>
> Thank you for your detailed feedback and for raising these concerns. We appreciate your constructive criticism and have addressed each of your points below:
>
> - **Regarding your concerns of novelty:**
>
>     We acknowledge that our contributions may appear incremental when viewed from a purely architectural standpoint. However, the main focus of our paper is not to propose a brand-new architecture, but rather to provide a comprehensive investigation of existing ideas around modeling directionality and demonstrate their practical utility in the context of directed link prediction.
>
>     Our work is intended to serve as a practical guide for practitioners by illustrating how simple, directed variants of existing techniques can enhance performance in directed settings. The directed distance encoding and the use of directed GNNs, while seemingly minor modifications, have demonstrated substantial impact on specific tasks, as evidenced by our results. Furthermore, our edge-wise structural feature extraction, while related to the Buddy method, is designed specifically to capture directionality in graph structures, which is not the primary focus of prior work. Instead of proposing a more complex model, we deliberately chose to evaluate simple heuristics and their directed variants, as we believe that understanding these fundamental aspects can often provide more value in real-world scenarios compared to developing highly complex, specialized models.
>
> - **Regarding choice of baselines:**
>
>     We acknowledge that including more advanced directed GNNs as baselines could provide additional context and strengthen our conclusions. Our primary goal was to provide a focused and systematic investigation of simpler, well-known methods and their directed variants in order to establish a clear and interpretable understanding of the role of directionality in link prediction. By using foundational models such as GCN, GraphSage, and GAT, our intent was to isolate the impact of incorporating directionality and understand its influence before introducing more complex elements.
>
>     The choice to exclude more advanced directed GNNs like was driven by a desire to avoid adding too many complex factors that could potentially obscure the impact of directionality itself. Our aim was to provide a more generalizable understanding, especially for practitioners who might prefer a simple yet effective solution for their tasks. That said, we do agree that including a few more advanced directed GNN models could help illustrate how more sophisticated architectures fare in comparison to DirLP, and we will consider expanding our baseline comparisons to include these models in future work.
>
> - **Regarding scalability and experimenting on larger datasets:**
>
>     We appreciate your comment on the scalability of DirLP to larger datasets. Although OGB-citation2, being a bipartite graph, is not the perfect fit, we agree that including larger, directed datasets would help to demonstrate the practical applicability of our approach on a wider scale. While during training DirLP does not impose extra complexity, scalability is indeed a consideration for DirLP, as the preprocessing step that include calculation of structural edge features has a quadratic complexity. Having said that, feature extraction is a one time cost and cashing proved to be simply useful in our experiments. In light of your review, we discuss potential avenues to further improve the efficiency of our approach in our conclusion as future work. However, given the time limitations, setting up an experimental evaluation for larger datasets is currently not very feasible.
>
> - **Additional Notes:**
>     - Regarding the Chameleon and Squirrel datasets, we appreciate the reference to potential issues of repeated edges leading to data leakage. In our experiments we used coalesced set of edges to avoid that.
>     - We understand your concern that the observed improvements in directed settings may seem trivial. However, our intention is to systematically investigate and demonstrate the extent to which directionality contributes to performance improvements in link prediction tasks. The directed components provide a finer granularity in capturing the asymmetrical relationships inherent in many real-world graphs, which we believe adds value beyond simple performance gains. The results serve to illustrate the importance of considering directionality explicitly, especially in applications where asymmetric relationships are critical.
>
> Once again, we appreciate your detailed feedback. We believe that the additional experiments, improved baselines, and efficiency considerations we plan to include in future work will significantly enhance the contributions of our study. Please let us know if you have any further suggestions or questions.

---

### Official Review · Reviewer_pwhV · 2024-10-23

**Soundness:** 3
**Presentation:** 4
**Contribution:** 3
**Rating:** 6
**Confidence:** 3

**Summary:**

The authors study the link prediction tasks in the directed settings. The authors conducted comprehensive experiments to compare the encoder, labeling tricks, structural features, decoder and negative sampling in both directed and undirected settings. Based on the observations from the experiments, the authors propose DirGNN targeting at performing link prediction tasks in the directed setting. Experiments across various benchmarks show that DirGNN can perform significantly better than the undirected sota methods.

**Strengths:**

[S1] The paper has a clear writing and easy to follow.

[S2] The authors conduct a comprehensive series of experiments on directed link prediction. It reveals which designs/components can improve the performance in the directed link prediction.

**Weaknesses:**

[W1] The benchmark datasets are small compared to most undirected link prediction methods. For example, OGB datasets, including Collab, PPA, and Citation2 are not used as benchmarks in this study. I would suggest including Citation2 as a benchmark since (1) it is large-scale, and (2) it is a directed graph dataset.

[W2] Directed GNN baselines are missing in the Table 6. The authors should include the popular directed GNNs (encoders) as baselines, like those discussed in the related work.

**Questions:**

[Q1] In Table 6, the authors can consider adding ablation studies to demonstrate how much each proposed directed component contributes to the overall performance improvement.

[Q2] The recent studies about link prediction methods focus on those that both have superior performance and capability to scale to large graphs. I wonder what extra computational cost DirGNN has compared to undirected GNNs.

---

> ### Author Response · Authors · 2024-11-22
> **Reply**
>
> Thank you for taking the time to review our work and provide valuable feedback. We appreciate your constructive comments and suggestions. Below, we address each of your points in detail:
>
> - **W1 - larger datasets:**
>
>     We understand your concern about the limited size of the benchmark datasets used in our study. Although OGB-Citation2, being a bipartite graph, is not the perfect fit, we agree that including larger, directed datasets would help to demonstrate the practical applicability of our approach on a wider scale. However, given the time limitations of rebuttal period, setting up an experimental evaluation for larger datasets is mentioned as future work in our conclusion.
>
> - **W2 - choice of baselines:**
>
>     We acknowledge that including more advanced directed GNNs as baselines could provide additional context and strengthen our conclusions. Our primary goal was to provide a focused and systematic investigation of simpler, well-known methods and their directed variants in order to establish a clear and interpretable understanding of the role of directionality in link prediction. By using foundational models such as GCN, GraphSage, and GAT, our intent was to isolate the impact of incorporating directionality and understand its influence before introducing more complex elements.
>
>     The choice to exclude more advanced directed GNNs was driven by a desire to avoid adding too many complex factors that could potentially obscure the impact of directionality itself. Our aim was to provide a more generalizable understanding, especially for practitioners who might prefer a simple yet effective solution for their tasks. That said, we do agree that including a few more advanced directed GNN models could help illustrate how more sophisticated architectures fare in comparison to DirLP, and we will consider expanding our baseline comparisons to include these models in future work.
>
> - **Q1 - ablation studies:**
>
>     Thank you for your suggestion. We believe, the experiments in Section 4  provide insights into the contributions of each component, and that adding further ablation studies could lead to overlapping findings. To avoid redundancy, we decided to focus on the existing experiments, which  showcase the impact of each directed component. We hope this clarifies our approach, and we are open to further suggestions if needed.
>
> - **Q2 - computational costs:**
>
>     Thank you for your question. In order to provide a sense of computational overhead, we have filtered 10 runs with the same number of layers and hidden dimensions for each type of encoder and calculated the mean and standard deviations of run durations per epoch:
>
>     | Dataset | Encoder | mean | std |
>     | --- | --- | --- | --- |
>     | Blog | DirGNN | 0.042640 | 0.009631 |
>     |  | GCN | 0.042439 | 0.006230 |
>     |  | GraphSage | 0.033597 | 0.005658 |
>     | Chameleon | DirGNN | 0.064700 | 0.013411 |
>     |  | GCN | 0.081281 | 0.009805 |
>     |  | GraphSage | 0.045391 | 0.005739 |
>     | CiteSeer | DirGNN | 0.117737 | 0.008673 |
>     |  | GCN | 0.119211 | 0.011380 |
>     |  | GraphSage | 0.101513 | 0.013186 |
>     | Cora | DirGNN | 0.097959 | 0.010057 |
>     |  | GCN | 0.069400 | 0.019150 |
>     |  | GraphSage | 0.061825 | 0.017471 |
>     | Squirrel | DirGNN | 0.154238 | 0.012540 |
>     |  | GCN | 0.157174 | 0.016864 |
>     |  | GraphSage | 0.142167 | 0.015270 |
>     | WikiCS | DirGNN | 0.141555 | 0.014427 |
>     |  | GCN | 0.172419 | 0.022934 |
>     |  | GraphSage | 0.127584 | 0.011555 |
>
>     From the results presented in the table, we observe that while there are some variations in computational costs between directed and undirected versions, these differences are not always consistent across datasets. In particular, DirGNN performs comparably to GCN and GraphSage in terms of runtime. However, the main overhead introduced by directionality in our experiments stems from the data preprocessing steps, which involve structural feature extraction with a quadratic complexity. This preprocessing is where the actual cost of introducing directionality becomes significant. Since it is a one-time cost, in practice, we found cashing the feature extraction output useful.
>
>
> Once again, we thank you for your insightful feedback. We believe that these additions will significantly enhance the comprehensiveness and clarity of our work, and we hope that our revisions address your concerns. Please let us know if you have further questions or suggestions.

---

> > ### Comment · Reviewer_pwhV · 2024-11-25
> >
> > Thanks for the responses. Given that OGBL datasets have been largely used as the benchmark for LP models, I still think it is necessary to include them, especially the directed OGBL-Citation2. Then it can prove what the authors claim, such that the directionality is beneficial and essential for LP tasks, particularly when compared to the undirected counterpart.

---

### Official Review · Reviewer_ye89 · 2024-10-27

**Soundness:** 2
**Presentation:** 2
**Contribution:** 2
**Rating:** 3
**Confidence:** 3

**Summary:**

The paper focuses on extending link prediction techniques to directed graphs, addressing a gap in Graph Representation Learning (GRL). While much of the research in link prediction has concentrated on undirected settings, the authors emphasize the importance of capturing directionality in many real-world applications (e.g., transaction networks). They propose **DirLP**, a novel framework for directed link prediction, which outperforms both heuristic-based and Graph Neural Network (GNN)-based models across multiple benchmark datasets. DirLP shows state-of-the-art performance across six datasets, proving its effectiveness in capturing the directionality of relationships.

**Strengths:**

1. The paper addresses a critical gap in GRL by extending established principles to directed settings.
2. The framework considers several key components such as asymmetric decoders, directed structural features, and labeling tricks to capture directionality effectively.The use of a modular design (e.g., DirGNN as encoder and CMLP as decoder) makes the framework adaptable to various graph structures.

3. Extensive experiments across multiple datasets (CORA, CITESEER, BLOG, etc.) show the model’s superiority, providing statistical insights into performance.

**Weaknesses:**

1. The necessity of directed setting is not clear. Commonly directed graphs are transformed to undirected graphs as the authors mentioned, so an important question is that can the model on directed setting perform better than undirected setting. The paper shows DirGNN can perform better than GCN or SAGE on directed graph, but how is it compared to the performances of GCN or SAGE on correspondent undirected graphs? So I think the comparison with GNNs on correspondent undirected graphs is important. For example, the authors can include a specific experiment comparing DirLP to GCN/GraphSAGE on both directed and undirected versions of the same datasets.

2. Large datasets from OGB are not included, which limits the application of proposed model. For example, the OGB-citation2 is a citation network which is obviously directed in raw data. Or can the authors discuss any potential scalability challenges DirLP might face on larger graphs?

3. What's $\alpha$ for each dataset? It's not shown in the hyperparameter table. Can the authors add it in Table 12 and discuss something on how the value of $\alpha$ show the different properties of each dataset?

**Questions:**

See weaknesses.

---

> ### Author Response · Authors · 2024-11-22
> **Reply**
>
> Thank you for your detailed feedback and for highlighting areas where our work could be made more robust. We appreciate your thoughtful comments and have addressed each of your points below.
>
> - **Regarding the role of directionality:**
>
>     Intuitively, directed edges allow us to differentiate a node's role as either a source or a target within a network, which provides finer granularity when modeling interactions. In many real-world contexts, this asymmetry is meaningful. For example, consider a transaction network where the goal is to identify fraudulent behavior. Suppose there are three types of participants: a parent, a teenager, and a fraudster. Let’s assume the model aims to understand the identity and behavior of the teenager. A money transfer of amount X from a parent to a teenager represents one type of pattern, while a transfer from a teenager to a fraudster represents another. If we look at a scenario where the teenager is receiving money from the parent, this pattern might be considered safe—such as receiving an allowance. However, if the teenager is instead sending the same amount to a fraudster, this could be a red flag. By taking into account the direction of these transactions, we can more effectively distinguish between benign and suspicious patterns. If we, instead, convert the originally directed network to an undirected one by symmetrizing the adjacency matrix, the sender/receiver roles of the teenager in these transactions would be lost, which may harm predictive performance in the aforementioned downstream task.
>
>     We appreciate you indicating your concern regarding this issue. To make the motivation of our work clearer, we have added an extended version of this example to our introduction.
>
>     Regarding the experiment you suggest on comparing models on undirected versions of datasets, we understand including such experiment would provide some insight but it could be outside the scope of this paper, as our primary focus is on understanding and evaluating the benefits of directionality in directed graphs. Having said that, a DirGNN is strictly proven to be more expressive than its undirected counterpart pf MPNN framework (See Theorem D11 in [1]).
>
>     [1] Rossi, E., Charpentier, B., Giovanni, F.D., Frasca, F., Günnemann, S., & Bronstein, M.M. (2023). Edge Directionality Improves Learning on Heterophilic Graphs. *ArXiv, abs/2305.10498*.
>
> - **Regarding scalability and experimenting on larger datasets:**
>
>     We appreciate your comment on the scalability of DirLP to larger datasets. Although OGB-citation2, being a bipartite graph, is not the perfect fit, we agree that including larger, directed datasets would help to demonstrate the practical applicability of our approach on a wider scale. While during training DirLP does not impose extra complexity, scalability is indeed a consideration for DirLP, as the preprocessing step that include calculation of structural edge features has a quadratic complexity. Having said that, feature extraction is a one time cost and cashing proved to be simply useful in our experiments. In light of your review, we discuss potential avenues to further improve the efficiency of our approach in our conclusion as future work. However, given the time limitations, setting up an experimental evaluation for larger datasets is currently not very feasible.
>
> - **Regarding the value of alpha in Equation 12:**
>
>     Thank you for pointing out the omission regarding the value of $\alpha$ for each dataset. In the revised version and we clarified that we set alpha to 0.5 for all datasets.
>
>
> Once again, we are grateful for your valuable feedback. We believe these additions will enhance the clarity and robustness of our work, and we hope our revisions address your concerns. Please let us know if you have any further questions or suggestions.

---

> > ### Comment · Reviewer_ye89 · 2024-11-25
> >
> > Thank you for your responses. I still think new experiments are needed for weakness 1 and 2 so I will keep my score.

---

### Official Review · Reviewer_bV2k · 2024-11-04

**Soundness:** 2
**Presentation:** 4
**Contribution:** 1
**Rating:** 3
**Confidence:** 4

**Summary:**

The authors propose a framework for understanding the directed link prediction problem with GNNs and how various changes to the underlying architecture can affect and then eventually enhance downstream performance. To test this, the authors run a series of experiments, both exploratory and ablative, to determine which practical components in directed link prediction architectures improve performance. After which, they construct their final framework to demonstrate it's superiority over common LP GNNs and heuristic methods.

**Strengths:**

* The intuition behind this problem is well-motivated, it is a common assumption within link prediction to assume that the input adjacency matrix is symmetric. Research, such as what the author's propose is necessary and potent for constructing GNN models that have more applicability in real-life scenarios where directed links and asymmetric adjacency matrices are more useful for end users.
* The authors integrate a variety of experiments in order to aid the reasoning for constructing their proposed framework. Since the design of these experiments is intuitive and easy-to-follow it makes the paper easier to read and provides clarity to the author's intent.
* Additionally, the scope of the methodology, including the questions it asks, is comprehensive in how it integrates reasoning at a scale that bridges a necessary gap between symmetric/undirected link prediction and directed link prediction, which is important for the proposed level of understanding that the authors are attempting to achieve.
* The use of a robust metric like MRR is an excellent practical decision and lends credence to the author's overarching claims about directed link prediction.

**Weaknesses:**

* There are concerns about intuition from the beginning, as seen in Figure 1. Why is it that practitioners should worry if the undirected or directed encoder captures varying levels of information? It is intuitive that more information can lead to better performance, especially given the experimental results, but the connection as to **why** the lack of directionality decreases performance is tantamount to improving understanding. For example, what can finer-levels of directed distance-encoding do in order to improve performance? Is there a stronger correlation between better expressiveness in directed networks and their performance than what is seen with undirected networks?
* The overlap in performance for many of the experiments is concerning, in order to determine the significance of whether or not varied encoders, decoders, labeling schemes are better than one another with confidence. Then p-values or interval tests should be conducted between the scoring distributions for all tested architectures to determine if there is an significant difference between the performance of these architectures. This is especially pertinent given the high-level of overlap between the reported scores in many of the experimental tables.
* The novelty of the framework is a concern given that the intent of this article is understanding how the design in directed link prediction can improve performance in more practical scenarios. What sort of insights does this paper provide that are not already evident within [1]? It seems that further investigation into directed distance encoding could provide a new level of insight.
* Although the overall scope of the preliminary experiments and ablation studies is comprehensive, the authors only tests the simpler and well-known baselines (heuristics, MLP, GAT, GCN, GraphSage) against DirLP. Why not include experiments that consider more advanced directed GNNs like MagNet [2] or DiGCN [3]? Or even additional directed autoencoders, given the preliminary questions explored in this article? The results about design principles remain inconclusive relative to integration of edge-wise structural feature extraction within DirLP without these sorts of considerations given that concepts which have been explored before in the directed GNN literature are not explored within this article.
* I am certain that the authors are aware of this. But, there are numerous spelling mistakes in the article, including the use of 'principal' instead of 'principle'.
* What search space of hyperparameters was OPTUNA given for tuning the tested models? This is important for experimental reproduction, but not critical given the Tables 9-12.

[1] Rossi, E., Charpentier, B., Giovanni, F.D., Frasca, F., Günnemann, S. &amp; Bronstein, M.M.. (2024). Edge Directionality Improves Learning on Heterophilic Graphs. Proceedings of the Second Learning on Graphs Conference in Proceedings of Machine Learning Research
[2] Xitong Zhang, Yixuan He, Nathan Brugnone, Michael Perlmutter, and Matthew Hirn. Magnet: A neural network for directed graphs. In NeurIPS, 2021
[3] Zekun Tong, Yuxuan Liang, Changsheng Sun, Xinke Li, David Rosenblum, and Andrew Lim. Digraph inception convolutional networks. In NeurIPS, 2020.
[4] Salha, G., Limnios, S., Hennequin, R., Tran, V. A., & Vazirgiannis, M. (2019, November). Gravity-inspired graph autoencoders for directed link prediction. In Proceedings of the 28th ACM international conference on information and knowledge management (pp. 589-598).

**Questions:**

See 'Weaknesses' section of this review for questions.

---

> ### Author Response · Authors · 2024-11-22
> **Reply**
>
> Thank you for your review and feedback. We appreciate the time and effort you took to engage with our work and provide insightful comments. Your suggestions have helped us improve both the clarity and depth of our work, and we are committed to addressing these points in our revision.
>
> - **Regarding the role of directionality:**
>
>     Directed edges differentiate a node's role as either a source or target, providing finer granularity when modeling interactions. In many real-world contexts, this asymmetry is meaningful. For example, consider a transaction network with participants such as a parent, a teenager, and a fraudster. A money transfer from a parent to a teenager is different from a transfer from a teenager to a fraudster. Receiving money from a parent might be considered safe, while sending the same amount to a fraudster could be a red flag. By considering transaction direction, we can better distinguish between benign and suspicious activities. If the directed network is symmetrized, the sender/receiver roles are lost, potentially harming predictive performance. We have added an extended version of this example to our introduction to clarify the motivation for our work.
>
> - **Regarding statistical significance tests:**
>
>     We agree that demonstrating statistical significance would enhance our findings. Initially, we reported the mean and standard deviation of MRR values over five random seeds, but individual data points were not stored, preventing direct statistical tests on MRR. Instead, we performed Student's t-tests on validation losses for different encoders and sampling strategies:
>
>     | Dataset | Encoders | t-value | p-value |
>     | --- | --- | --- | --- |
>     | Cora | GCN vs. GraphSage | 1.959 | 0.066 |
>     | Cora | GCN vs. DirGNN | 13.118 | 1.19e-10 |
>     | Cora | GraphSage vs. DirGNN | 8.567 | 9.11e-08 |
>     | Chameleon | GCN vs. GraphSage | 1.747 | 0.104 |
>     | Chameleon | GCN vs. DirGNN | 4.418 | 0.0003 |
>     | Chameleon | GraphSage vs. DirGNN | 1.384 | 0.190 |
>     | Blog | GCN vs. GraphSage | 1.148 | 0.284 |
>     | Blog | GCN vs. DirGNN | -0.490 | 0.637 |
>     | Blog | GraphSage vs. DirGNN | -0.676 | 0.518 |
>
>     The results show that DirGNN significantly outperforms GCN on Cora and Chameleon. For sampling strategies, there is no statistically significant difference:
>
>     | Dataset | t-value | p-value |
>     | --- | --- | --- |
>     | Cora | -1.381 | 0.184 |
>     | Chameleon | -0.321 | 0.754 |
>     | Blog | 1.244 | 0.235 |
>
>     We aim to repeat our experiments to test performance in terms of MRR directly, though the rebuttal period may not be sufficient for this.
>
> - **Regarding novelty:**
>
>     While our contributions may appear incremental, our main focus is to conduct a comprehensive investigation of modeling directionality and demonstrate its utility in directed link prediction. Our work serves as a practical guide for practitioners by showing how simple, directed variants of existing techniques can enhance performance. Directed distance encoding and the use of directed GNNs, while seemingly minor, have a substantial impact on specific tasks. Instead of proposing a complex model, we chose to evaluate simple heuristics and their directed variants, believing that understanding these fundamental aspects is often more valuable in real-world scenarios.
>
> - **Regarding choice of baselines:**
>
>     Including more advanced directed GNNs as baselines could strengthen our conclusions. Our goal was to systematically investigate simpler, well-known methods and their directed variants to establish a clear understanding of directionality's role in link prediction. By using foundational models like GCN, GraphSage, and GAT, we aimed to isolate the impact of directionality before introducing more complexity. In future work, we plan to include more advanced directed GNNs to provide a broader comparison.
>
> - **Additional Notes:**
>
>     Thank you for pointing out the typos. We will fix these in our revised version. Additionally, we have added our search space parameters as an appendix.

---

> ### Comment · Reviewer_bV2k · 2024-11-24
>
> Thank you for your rebuttal, I agree that quantification of these problems is important for enhancing understanding; the likes of which can then inspire future research. I have read the rebuttal and edits in the paper. The additional details and definitions enhance the understanding and purpose behind this paper. I especially appreciate the statistical significance tests on validation loss, such a test seems far more practical than testing the scoring distributions directly.
>
> Despite this, I still have concerns:
> * This paper does an excellent job of quantifying the problem and what components of the DirLP framework improve performance over undirected methods. I appreciate the explanation on the *why*, I am still not sure *what* exactly the DirLP framework provides over DirGNN alone. The trade-offs between computation time of the additional components in the DirLP framework coupled with the results between Table 1 and Table 6 seems to indicate that any improvement to DirGNN is marginal. This is especially concerning given that DirGNN or any other model designed specifically for directed link prediction is not included in Table 6 results.
> * I am assuming that the variations of GCN and GraphSAGE tested in Table 6 are not the same variations tested in Table 1, even though the results are nearly identical for the Cora dataset between both tables? Is this a typo? It is not clear which variation of GAT, GCN, and GraphSAGE are tested within Table 1 and Table 6, are the variations tested directed in both?
> * If this paper is to serve as a guide for designing directed link prediction frameworks, then why not enhance methods such as GCN and GraphSAGE within the DirLP framework (if possible)? Is there any way to make a conclusion on what components within the DirLP frameworks that users might prioritize in order to squeeze out the most performance in directed link prediction?
> * The revised paper extends beyond the [10-page limit set by the ICLR conference](https://www.iclr.cc/Conferences/2025/AuthorGuide). Although the new content in the paper is helpful, editing the text or placing few paragraphs in the appendix is necessary in order to meet conference requirements.
> * I encourage the authors to highlight any edits or addtions made to the paper in either red or blue to clearly indicate any updates made to the paper.
>
> Due to these concerns, I will maintain my current score.

---

### Meta-Review · Area_Chair_28nt · 2024-12-17

**Metareview:**

The paper proposes a framework for directed link prediction, focusing on the impact of directed components (e.g., DirGNN and directed distance encoding) on performance. Several reviewers noted that the paper lacks novelty, with the directed components offering only incremental improvements over existing methods. Concerns were raised about the use of small benchmark datasets and the absence of larger datasets like OGB, which are crucial for validating the generalizability of the approach. Additionally, issues such as potential data leakage from repeated edges in the Chameleon and Squirrel datasets, and high computational complexity (e.g., the O(N³) labeling trick) were highlighted. Although the authors indicated plans to include more baselines and scale their experiments in future work, these issues were not adequately addressed in the current version. Overall, the paper provides valuable insights but lacks sufficient innovation and experimental support for publication.

**Additional Comments On Reviewer Discussion:**

Although the authors responded, they did not provide additional experiments to address the related issues.

---

### Decision · Program_Chairs · 2025-01-22

Reject